# Expanding the Pathogen Panel in Wastewater Epidemiology to Influenza and Norovirus

**DOI:** 10.3390/v15020263

**Published:** 2023-01-17

**Authors:** Rudolf Markt, Fabian Stillebacher, Fabiana Nägele, Anna Kammerer, Nico Peer, Maria Payr, Christoph Scheffknecht, Silvina Dria, Simon Draxl-Weiskopf, Markus Mayr, Wolfgang Rauch, Norbert Kreuzinger, Lukas Rainer, Florian Bachner, Martin Zuba, Herwig Ostermann, Nina Lackner, Heribert Insam, Andreas Otto Wagner

**Affiliations:** 1Department of Microbiology, Universität Innsbruck, 6020 Innsbruck, Austria; 2Department of Health Sciences and Social Work, Carinthia University of Applied Sciences, 9020 Klagenfurt, Austria; 3Institut für Umwelt und Lebensmittelsicherheit des Landes Vorarlberg, 6900 Bregenz, Austria; 4Department of Infrastructure, Universität Innsbruck, 6020 Innsbruck, Austria; 5Institute for Water Quality and Resource Management, Technische Universität Wien, 1040 Vienna, Austria; 6Austrian National Public Health Institute, 1010 Vienna, Austria

**Keywords:** wastewater-based epidemiology, influenza, norovirus, surveillance, digital PCR, PEG-precipitation, wastewater, inpatients

## Abstract

Since the start of the 2019 pandemic, wastewater-based epidemiology (WBE) has proven to be a valuable tool for monitoring the prevalence of SARS-CoV-2. With methods and infrastructure being settled, it is time to expand the potential of this tool to a wider range of pathogens. We used over 500 archived RNA extracts from a WBE program for SARS-CoV-2 surveillance to monitor wastewater from 11 treatment plants for the presence of influenza and norovirus twice a week during the winter season of 2021/2022. Extracts were analyzed via digital PCR for influenza A, influenza B, norovirus GI, and norovirus GII. Resulting viral loads were normalized on the basis of NH_4_-N. Our results show a good applicability of ammonia-normalization to compare different wastewater treatment plants. Extracts originally prepared for SARS-CoV-2 surveillance contained sufficient genomic material to monitor influenza A, norovirus GI, and GII. Viral loads of influenza A and norovirus GII in wastewater correlated with numbers from infected inpatients. Further, SARS-CoV-2 related non-pharmaceutical interventions affected subsequent changes in viral loads of both pathogens. In conclusion, the expansion of existing WBE surveillance programs to include additional pathogens besides SARS-CoV-2 offers a valuable and cost-efficient possibility to gain public health information.

## 1. Introduction

Wastewater-based epidemiology (WBE) is a powerful tool for monitoring the spread of viral pathogens within a population connected to a sewage system, as shown during the COVID-19 pandemic [1,2,3,4,5]. Viruses such as poliovirus [6,7], hepatitis E virus [8], norovirus (NoV) [9,10], influenza virus [11,12], and many more have already been detected [13] and monitored in wastewater. In contrast to traditional individual-level testing, WBE offers several advantages for the cost-efficient early detection of local or regional outbreaks and long-term monitoring of pathogens at community level [14]. Hence, comparability among different wastewater treatment plants (WWTPs) is essential and can be achieved by using population size markers to normalize WBE data [15,16]. In this context, NH_4_-N concentration or NH_4_-N load is a valuable population size marker [16,17,18]. This allows an efficient pandemic or pathogen management, including the implementation and validation of non-pharmaceutical interventions (NPIs) [18]. Recently, the COVID-19 pandemic demonstrated the importance of WBE for pathogen surveillance and accelerated the world-wide implementation of this tool [19].

Although WBE is a cost-efficient way to track the spread of pathogens at community-level if calculated per single community member, the per-sample costs can be high. Sample logistics significantly contribute to the high costs, especially in rural regions [20]. Currently, wastewater samples are mainly prepared for the subsequent quantification of SARS-CoV-2. During this process, the genomic material of many viruses extracted as by-catch along with SARS-CoV-2 RNA is usually disregarded, although it could contain valuable information.

Influenza virus causes a seasonal epidemic (the flu season). The enveloped virus is usually spread through respiratory droplets and aerosols. Influenza A virus (IAV), influenza B virus (IBV), and influenza C virus infect humans, with IAV and IBV being responsible for most cases. IAV is normally responsible for epidemics and pandemics. IBV undergoes genetic and antigenic changes to a smaller extent and shows increased activity every 2–4 years [21]. Worldwide, influenza viruses result in up to five million cases of severe illness every year [22]. In the Northern Hemisphere, the typical flu season spans from October to May. The virus is transmitted mainly through the respiratory path, but its RNA may also be detected in the stool of symptomatic patients, with a detection rate ranging from 7.2% to 47% [23,24,25,26].

Noroviruses (NoVs) are a common cause of gastroenteritis and are usually spread through the fecal–oral route [27]. Several genogroups of the non-enveloped NoVs can be distinguished. The genogroups GI and GII are the most relevant to humans and are responsible for local and regional outbreaks. NoVs are a highly contagious group of viruses that have resulted in nearly 700 million annual cases worldwide [28], with infection commonly occurring during the winter months. The RNA of NoVs can be detected in the stool of patients with gastroenteritis with a detection rate ranging from 17.9% to 38.3% [29,30]. This RNA can still be detected several weeks after infection [31]. Another study found NoV in stool samples of 361 out of 2205 (16.4%) asymptomatic individuals [32].

Therefore, the aims of this retrospective study were (I) to detect and quantify two different viruses from wastewater extract as a by-catch of a WBE-based SARS-CoV-2 surveillance program, applying NH_4_-N as a population marker; (II) to compare the WBE data with incidence from inpatients; and (III) to examine the effect of COVID-19-related NPIs on the spread of the other pathogens. For this purpose, we searched for the enveloped, respiratory IAV and IBV known for their droplet-based transmission, and the non-enveloped, gastro-enteric NoV of GI and GII known for their fecal–oral transmission in already present wastewater extracts. The study includes samples from the SARS-CoV-2 surveillance program in Austria from the winter season of 2021/2022 [33].

## 2. Materials and Methods

### 2.1. Sample Collection and Sample Preparation

Approximately 250 mL of 24 h volume-proportionally-mixed influent was collected twice a week from 11 WWTPs in two federal states of Austria from September 2021 to April 2022 (Figure 1). Five WWTPs in Vorarlberg (V001–V005) covering ~62% of the population and six WWTPs in Salzburg (Z001, Z002, Z003, Z006, Z008, Z012) covering ~80% of the population were monitored. Raw samples were chilled to 4 °C [34] and processed within two days. Concentration via polyethylene glycol precipitation (PEG precipitation) and purification via the Monarch™ Total RNA Miniprep Kit (New England Biolabs, Ipswich, MA, USA) were conducted according to Markt et al. (2021) [34]. Residual RNA extracts after SARS-CoV-2 surveillance were stored at −80 °C until one-step reverse-transcriptase digital PCR analysis (RT-dPCR). Daily wastewater flow rate and ammonium–nitrogen concentration (NH_4_-N measured using standard commercial cuvette tests) were reported by the WWTPs.

### 2.2. Probe Based RT-Digital PCR-Detection of Influenza Virus and NoV

For the detection of IAV, primers and probes targeted the matrix proteins 1 and 2 in segment 7, and for IBV the nuclear export protein (NEP) and nonstructural protein 1 (NS1) in segment 8 of the genome, according the SARS-CoV-2/IAV/IBV multiplex assay published by the US Centers for Disease Control and Prevention [35]. Primers and probes targeting the ORF1–ORF2 junction were used for specific detection of NoV GI and GII following a duplex approach [36]. All primers and probes were obtained from Microsynth AG (Balgach, Switzerland). Quenchers and reporters were modified. Details on Primers and Probes can be found in Appendix A. Positive control sequences (Appendix A) for IAV, IBV, NoV GI and NoV GII were obtained from Eurofins Genomics AT (Vienna, Austria) as plasmids cloned into a standard plasmid vector (pEX-A128) (Appendix A). In vitro amplification of the plasmid was performed using the NEBNext Ultra™ II Q5 Master Mix (New England Biolabs, Ipswich, MA, USA) using a standard vector primer pair (standard PEX primer) from Eurofins Genomics AT (Vienna, Austria). The thermocycling conditions for amplification of the positive controls were set according to the manufacturer’s protocol for the Q5 High-Fidelity DNA Polymerase. Agarose-gel purification of the resulting PCR product was performed using the Monarch DNA Gel Extraction Kit (New England Biolabs, Ipswich, MA, USA). Finally, the concentration of the in vitro amplified linear DNA was measured using a Quantus Fluorometer E6150 (Promega, Madison, WI, USA) and diluted to <1000 gcµL^−1^. The PCR products were used as positive controls and were integrated into the assay as linear dsDNA. To detect and quantify absolute RNA copy numbers, two duplex RT-digital PCR (RT-dPCR) assays were performed on a QIAcuity One 5-plex digital PCR system (QIAGEN GmbH, Hilden, Germany): one for influenza viruses and one for NoV detection. RT-dPCR of the 20 µL reaction mixture consisted of: 10 µL of Luna Universal Probe One-Step Reaction Mix (2x); 1 µL of Luna WarmStart RT Enzyme Mix (20x) (both New England Biolabs, Ipswich, MA, USA); 3.2 μL of mixed primers (forward and reverse, final concentration 0.8 μM for each target); 0.8 μL of mixed probes (final concentration 0.2 μM for each target); 0.5 µL of reference dye, and 4.5 μL of template. The PCR program was identical for both assays: An initial reverse transcription step at 55 °C for 10 min was followed by initial denaturation at 95 °C for 1 min. Subsequently, a total of 40 cycles at 95 °C for 10 s and 60 °C for 40 s were performed within 96-well Nanoplates with 8500 partitions per well (QIAGEN GmbH, Hilden, Germany). Fluorescence was read at the end of the 40 cycles. Gene copy numbers per positive partition were calculated according to the Poisson distribution. The concentration in the reaction mix was calculated based on the volume of positive and negative valid partitions. These steps were done using the internal software Qiacuity Software Suite (QIAGEN GmbH, Hilden, Germany). Subsequently, the results were compensated for the volume of the template in the dPCR reaction, the RNA extract eluted after purification, and the initial wastewater sample volume. The limit of detection for InfA and InfB was 1 gc per reaction, and for NoV GI and GII 3 gc per reaction.

### 2.3. Inpatients Infected with NoV or Influenza Virus and Anti-COVID-19 NPIs

Inpatients with primary or secondary infection with influenza virus (J09, J090, J091, J098, J10, J100, J101, J101, and J108 according to ICD-10-GM-2022) and norovirus (A081 according to ICD-10-GM-2022) were aggregated according to admission week and federal state. Six events increasing or decreasing the impact of NPIs on the spread of the COVID-19 pandemic were set from September 2021 to March 2022: (I) September 9—school opens after the summer holidays, (II) November 15—lockdown for the unvaccinated, (III) November 22nd—complete lockdown, (IV) December 12—partial opening, (V) January 11—Omicron measures, and (VI) March 5—complete opening.

### 2.4. Normalization of Wastewater Signal and Statistical Analysis

Viral load was defined as the concentration of gene copies detected multiplied by the daily wastewater flow rate. For comparison between WWTPs, viral loads were normalized to NH_4_-N load assuming a NH_4_-N excretion per person and day of 8 g N [37,38] resulting in mega (M), giga (G), or tera (T) gene copies per person per day (gcPE^−1^/d^−1^). To test for significant difference of reported NH_4_-N concentration between WWTPs, the Kruskal–Wallis rank sum test and the Wilcoxon rank sum as post hoc test with Bonferroni correction were performed.

Wastewater data were down-sampled from biweekly datapoints to weekly datapoints (mean loads of both datapoints), as inpatient numbers were resolved at a weekly basis [39]. To compare wastewater signal and inpatient number, a cross-correlation analysis and ranked correlation analysis (Spearman rank correlation test) between aggregated viral loads (sum of viral loads) per calendar week across all WWTPs and inpatient numbers were performed. The significance cut-off was set at α = 0.05 and at α = 0.005 for highly significant for all analyses. Smoothing was conducted using a LOWESS-smoothing with a span of 0.5.

Calculations, statistical analysis, and figures were prepared in RStudio© (2022.07.1 + 554) with R version 4.1.1. Data organization, normalization, and statistical analysis were carried out using R-base [40]. Further data organization was carried out using the packages tidyquant [41] and readxl [42]. Figures were prepared using the packages ggplot2 [43] and cowplot [44].

## 3. Results

Approximately 70% of the total population of the two Federal States Salzburg and Vorarlberg, or 0.7 Mio. inhabitants, are connected to the sewage system of the 11 WWTPs monitored for this study. The number of connected inhabitants per WWTP ranged from 321,000 (Z001) to 14,000 (Z012) as shown in Table 1. Compared with registered numbers, the population size marker NH_4_-N slightly overestimated the number of inhabitants with 113% ± 25% (mean ± SD) across all WWTPs. A conspicuous discrepancy was found for the WWTPs V003 (146%) and Z002 (170%). Moreover, highly significant differences were detected in the NH_4_-N concentration between WWTPs. The post hoc test showed that WWTPs Z002 and Z008 differed significantly from other WWTPs. Wastewater flow rate, NH_4_-N concentration, and the calculated population for each of the WWTPs are depicted in Appendix A.

In total we analyzed 538 RNA extracts. IAV, NoV GI, and NoV GII were detected at least once in each of the WWTPs, while IBV was not detected at all between September 2021 to March 2022. The normalized IAV load (Figure 2A) was three orders of magnitude lower than the normalized NoV GII load (Figure 2B). IAV was under the limit of detection for many WWTPs during autumn, but started to rise in the middle of December. The IAV signal still rose at the end of the measurement period in March 2022. On the contrary, NoV load in wastewater was highest during autumn. Comparing the two NoV groups, NoV GII was ten to a hundred times more abundant than NoV GI. Moreover, NoV GI and NoV GII did not show the same temporal dynamics. NoV GI was highest in the middle of September (calendar week 38–39, 2021), while NoV GII peaked four to five weeks later (calendar week 43–44, 2021).

Inpatient numbers for both viral infections were low during the measurement period compared with previous years (Figure 3). Additionally, it was not possible to distinguish between virus variants or genogroups for influenza or NoV, respectively. Although resolution of the data allowed WWTP-specific inpatient numbers, the granularity was very high at this level. Furthermore, patients from rural regions may change catchment area when hospitalized. Therefore, inpatient numbers were aggregated at a level which includes both federal states, together covering approx. 0.95 Mio. inhabitants. The number of hospitalized persons infected with NoV peaked in autumn 2021 (Figure 4B). Distinct from that dynamic, the number of hospitalized persons infected with the influenza virus started to rise during the Christmas Holidays of 2021 and continued to rise at the end of the measurement period in March 2022 (Figure 4A).

Cross-correlation analysis showed a significant correlation between the aggregated virus load from all WWTPs and the aggregated inpatient numbers using IAV viral load (Appendix A) and NoV GII viral load (Appendix A) at a weekly basis. Neither IAV nor NoV GII showed a positive or negative lag in their temporal dynamics in comparison with the timeseries from respective inpatient numbers integrating the complete analysis period. Hence, the rank correlation test was also highly significant for IAV and significant for NoV GII. No significant correlation was obtained by including NoV GI load into analysis.

NPIs due to the COVID-19 pandemic affected the spread of influenza and NoV in Austria. The normally occurring seasonal peak during the winter period was missed in 2020. Additionally, the onset for the winter season 2021/2022 was earlier and the inpatient number was lower for influenza infections in comparison with seasons before the COVID-19 pandemic (Figure 3). Effects of NPIs on the spread of both of the investigated pathogens were complex. Nevertheless, it could be shown that a change in NPIs strategy coincided with a change of the virus signal in wastewater and the number of inpatients (Figure 4). Hence, in temporal proximity to the lockdowns (calendar week 46–47, 2021), virus loads for IAV and NoV GII decreased in November 2021. After Omicron-related measures (calendar week 02, 2022), the signal of NoV from wastewater dropped again in January 2021. The signal drop of IAV after Omicron-related measures was temporally shifted, highlighting the complexity of this topic. In contrast to restrictions, relaxation of regulations came along with an increase in virus loads as can be seen for the partial opening (calendar week 50, 2021) in December 2021 and the complete opening (calendar week 09, 2022) in March 2022.

## 4. Discussion

Archived RNA extracts from SARS-CoV-2 surveillance could easily be used to retrospectively track the seasonal occurrence of infections with influenza and norovirus. To normalize pathogen concentration, we assumed a daily NH_4_-N load per person of 8 gN [37] and were able to emulate the number of inhabitants attached to the sewer system by 113% ± 25%. The conspicuous result for WWTP Z002 (overestimation of population by 170%) may be explained by a significantly high NH_4_-N concentration, causing an unexpectedly high daily NH_4_-N load. Investigations for WWTP Z002 revealed that increased NH_4_-N concentrations came from internal returns of process water (anaerobic sludge desiccation). NH_4_-N concentration of the second conspicuous WWTP, V003 (overestimation of population by 146%), was not significantly different from other WWTPs assuming normal, urban wastewater. However, investigations for WWTP V003 revealed that industrial discharges (from an industrial bakery and industrial meat processors) as well as internal returns of process water (anaerobic sludge desiccation) increased the NH_4_-N load. Consequently, we recommend normalization of viral loads for inter-WWTP comparability, but the applicability of the population-size marker NH_4_-N must be verified for each WWTP.

Non-pharmaceutical interventions during the COVID-19 pandemic affected the spread of influenza viruses [45,46] and NoV [47,48]. In comparison with the influenza and NoV epidemics of 2017–2019, the annual sharp increase in infected inpatients was missing for both pathogens in 2020 and 2021. In our study, IAV and NoV showed different prevalence, with concentrations of NoV being approximately 1000 times higher. IBV activity was low during the winter season 2021/2022 in the northern hemisphere [49] and was not found in wastewater samples. Although, IAV activity was low during the measurement period due to anti-COVID measures, quantifiable traces of genomic material were found in wastewater via RT-dPCR. Aside from low prevalence, several other factors could explain the low IAV concentration in wastewater. These include the droplet-based transmission route via the respiratory tract, the relatively low fecal shedding [24,25,26], and the lower stability of enveloped compared to non-enveloped viruses in wastewater [13]. Additionally, recent studies reported that IAV is predominantly found in the solid fraction of wastewater [11,12]. This fraction was neglected during sample preparation for this study, as the main aim of RNA extraction was the detection of SARS-CoV-2 genomic material. According to our experience, the purification of genomic material of SARS-CoV-2 from wastewater is more easily achieved by processing the liquid fraction. Nevertheless, the signal from the liquid fraction would be sufficient for a surveillance program regarding the spread of IAV. In contrast to IAV, the non-enveloped NoV causes waterborne outbreaks, is shed via feces in large quantities, and is relatively stable in wastewater [50], and thus is suitable for WBE surveillance from already existing extracts.

Correlation between viral loads in wastewater and inpatients were significant for both, IAV and NoV GII, which is in accordance with previous studies [9,10,11,12]. Inclusion of the GI wastewater signal reduced the correlation coefficient which indicates a difference in hospitalization rates between GII and GI. Clinical information on the matter, however, is sparsely available. Desai et al. (2012) [51] report in their review including 71,724 illnesses and 501 hospitalizations a significantly higher hospitalization rate in GII.4 cases than in non-GII.4 cases, showing that different genotypes can generally lead to different case severity. Cross-correlation analysis indicated that the wastewater signal coincided with inpatient numbers for IAV und NoV GII. Huang et al. (2022) [10] reported similar results for NoV GII comparing wastewater data with cases and outbreaks of acute gastroenteritis. Kazama et al. (2017) [9] reported similar results for NoV in comparison with clinical cases.

In contrast to our findings, Mercier et al. (2022) [11] reported a wastewater lead time for influenza of more than two weeks over inpatient testing and testing of institutional outbreaks. Discrepancies between our study and other findings regarding lead time may have several causes. One important aspect is that in our study infected inpatients were assigned to the admission week and not to the date of onset of symptoms or diagnosis. An additional limitation is the temporal resolution at a weekly basis which may be too coarse for lead/lag-time calculations for periods smaller than seven days. Concerning NoV, we observed further an interesting temporal shift of four to five weeks between NoV GI and GII. NoV activity normally increases during the cold months, with lower activity in summer [52]. However, NoV GI concentration was highest in September in the present study. Nordgren et al. (2009) reported a high concentration of NoV GI in wastewater during the summer in contrast to NoV GII [53]. Thus, NoV surveillance should also consider the warmer seasons.

To prove a causal relationship between single NPIs and pathogen spread, additional variables including measures-fatigue, NPI-awareness due to medial impact, weather conditions, and mobility would be necessary. Nevertheless, we were able to assign the onset of restrictive measures to a subsequent decrease and the relaxation of measures to a subsequent increase in wastewater signal for both IAV and NoV GII.

## 5. Conclusions

It could be shown that an existing WBE program for SARS-CoV-2 surveillance can easily be expanded to monitor influenza and norovirus. Concentrated and purified RNA extracts from the liquid fraction of wastewater for SARS-CoV-2 detection contains sufficient genomic material for the surveillance of both influenza virus and norovirus. The resulting data contain valuable public health information on the spread of viral pathogens. Further, normalization for population size is independent from the number of pathogens tracked. Thus, the effort for normalization remains constant, while the benefit increases with every additional pathogen detected. The molecular biological detection of additional pathogens as by-catch in existing WBE infrastructure offers great potential for pathogen surveillance.

## Figures and Tables

**Figure 1 viruses-15-00263-f001:**
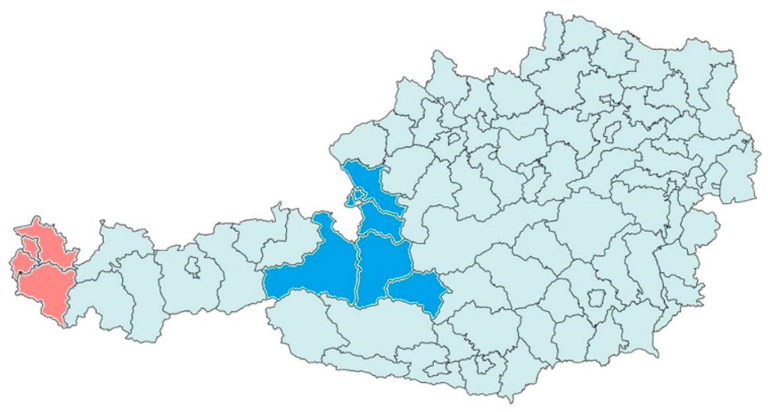
Map of Austria and its political districts. Colored districts correspond to those containing the 6 WWTPs monitored in Salzburg (blue) and the 5 WWTPs in Vorarlberg (red), covering in total 0.7 Mio inhabitants.

**Figure 2 viruses-15-00263-f002:**
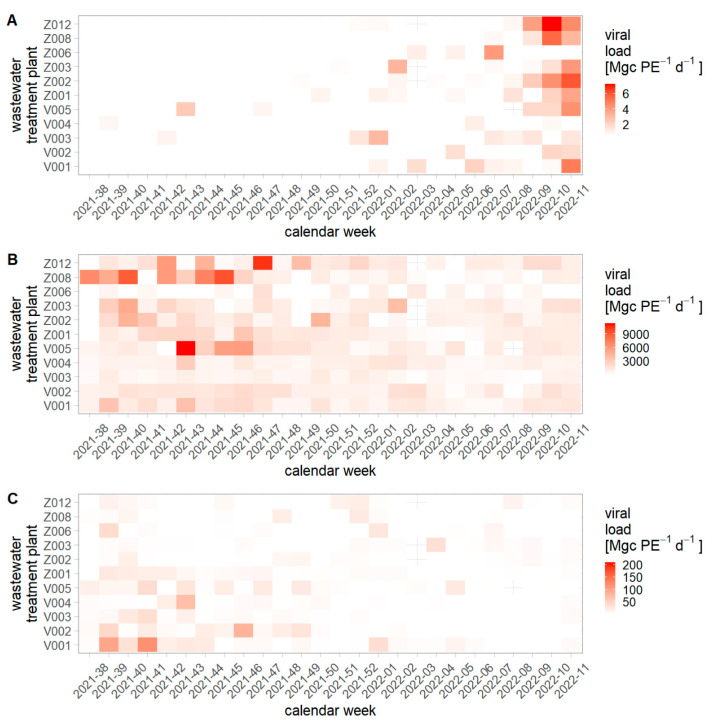
Normalized virus loads for (**A**) influenza A, (**B**) norovirus GII, and (**C**) norovirus GI per WWTP. More intense color corresponds to higher loads in wastewater. Each plot is scaled separately due to large discrepancy of absolute loads between pathogens. WWTP V001–V005 correspond to the federal state Vorarlberg and WWTPs Z001–Z012 correspond to the federal state Salzburg.

**Figure 3 viruses-15-00263-f003:**
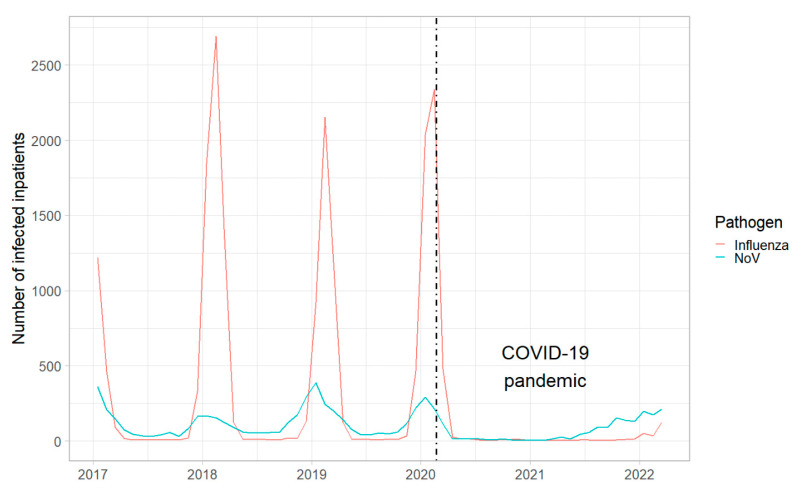
Seasonal occurrence of (red line) influenza virus and (blue line) norovirus infections in inpatients aggregated per month across Austria. The dashed black line marks the date of the first clinical COVID-19 case in Austria.

**Figure 4 viruses-15-00263-f004:**
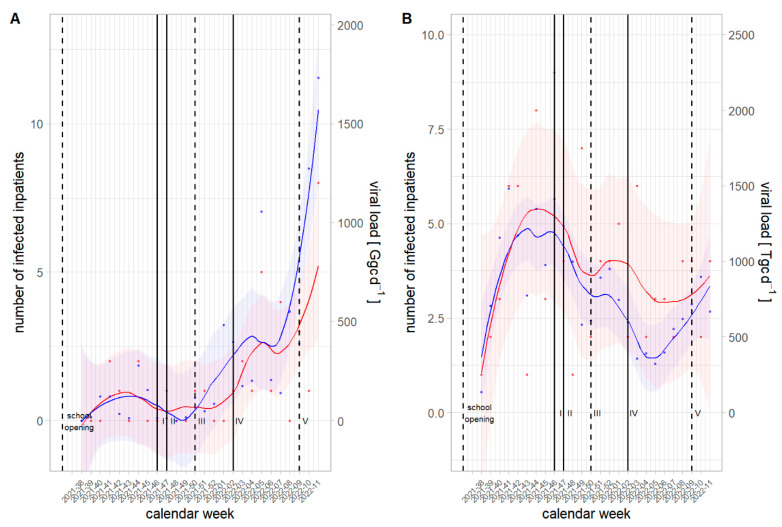
Comparison between viral loads in wastewater and number of infected inpatients from September 2021–March 2022 including time points for non-pharmaceutical intervention for IAV (**A**) and NoV GII (**B**). (Blue dots) Viral loads and (red dots) number of inpatients were aggregated at a weekly basis. Smoothing lines with 95% confidence bands correspond to LOWESS-smoothing. Six events increasing or decreasing the impact of NPIs on the spread of the COVID-19 pandemic were set from September 2021 to March 2022: September 9–school opens, (I) November 15—lockdown for those unvaccinated, (II) November 22nd—complete lockdown, (III) December 12—partial opening, (IV) January 11—Omicron measures, (V) March 5—complete opening.

**Table 1 viruses-15-00263-t001:** WWTPs monitored in this study including relative coverage corresponding to total population within the two federal states Vorarlberg and Salzburg, and the number of inhabitants connected to each sewage system. Calculated populations across all sampling points per WWTPs are stated as theoretical inhabitants after NH_4_-N normalization. Additionally, the ratio between theoretical and registered inhabitants was calculated in %.

Federal State (Population Covered by the WWTPs Monitored)	WWTP ID	Inhabitants Connected According Civil Register	Theoretical Population Calculated by NH_4_-N-Normalization (Mean ± SD)	Calculated Population/ Registered Population
Vorarlberg (~62% coverage)	V001	66,961	69,363 ± 13,908	104%
	V002	57,465	62,984 ± 13,152	110%
	V003	51,366	75,197 ± 14,846	146%
	V004	41,696	44,686 ± 8655	107%
	V005	32,471	32,373 ± 11,170	100%
Salzburg (~80% coverage)	Z001	320,682	273,966 ± 41,851	85%
	Z002	38,606	65,808 ± 10,666	170%
	Z003	31,638	38,317 ± 10,562	121%
	Z006	22,447	22,522 ± 3721	100%
	Z008	20,370	16,439 ± 1749	81%
	Z012	14,200	16,491 ± 4078	116%

## Data Availability

Data can be shared on request.

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
