# Peer review of "Expanding the Pathogen Panel in Wastewater Epidemiology to Influenza and Norovirus"

_viruses, 2023, doi:10.3390/v15020263_

Round 1

Author Response

The authors offer great thanks to the reviewer who took the time to add revisions and good suggestions. Please see the attachment for specific responses. We hope that these responses satisfy the reviewers’ requests.

Reviewer 2 Report

The manuscript by Rudolf Markt et al., describes the “Expanding the pathogen panel in wastewater epidemiology to influenza and norovirus”. It’s a retrospective study of wastewater-based pathogens epidemiology. The authors collected wastewater specimens biweekly from 11 treatment plants in 2 federal states for the purpose of monitor SARS-CoV 2 surveillance in September 2011-April 2022.

Those specimens were expanded screening viral loads of influenza A, B, norovirus genogroups GI and GII by digital RT-PCR. The viral loads were compared with incidence of inpatient.

Comments:

Norovirus is a leading cause of acute diarrhea disease of all population in the world. Cases of norovirus infected through fecal-oral and were with/without symptoms all year round but peak most occurred in low-temperature season. Influenza virus is transmitted through the respiratory pathway and causes illness same in winter and spring. This article is a good idea that expanding wastewater-based monitor in other viruses and provided comprehensive analysis. Though, there were a couple of things that would improve the paper quite a bit, including the following critical points.

1.      The detection limitation will influence the sensitivity of pathogens in monitor system. Please provide the detection limit of each virus digital RT-PCR in this article.

2.      Influenza virus most secret and transmit by respiratory pathway only a few manuscripts indicate detected RNA in stool from case with symptom. The low viral load in wastewater might influence the long-term detection or prediction as shown in Fig2 and Fig 4.

3.      Most of norovirus or influenza virus infect cases are without or with mild symptom.

The information of pathogens detection from wastewater should earlier than increase numbers of inpatient.

In this study, viral load data were compared with infected cases number of inpatient. Please explain why viral loads were compare with infected cases number of inpatient not outpatient.

In the study period, there were 6 NPIs to control the spread of pandemic SARS-CoV 2.  In Fig2 and Fig 4, indicate not all lockdown NPIs policy or measure have the same impact on norovirus or influenza virus, such as after Jan 11th the numbers of IAV inpatients increase.

Author Response

(The authors gave the same response as above.)
